# A Novel Model for the RNase MRP-Induced Switch between the Formation of Different Forms of 5.8S rRNA

**DOI:** 10.3390/ijms22136690

**Published:** 2021-06-22

**Authors:** Xiao Li, Janice M. Zengel, Lasse Lindahl

**Affiliations:** 1Department of Biological Sciences, University of Maryland Baltimore County (UMBC), 1000 Hilltop Circle, Baltimore, MD 21250, USA; xiao.li@abbott.com (X.L.); Janicezengel@gmail.com (J.M.Z.); 2Department of Biology, University of Rochester, Rochester, NY 14627, USA

**Keywords:** ribosome biogenesis, rRNA processing, RNase MRP, long/short 5.8S rRNA

## Abstract

Processing of the RNA polymerase I pre-rRNA transcript into the mature 18S, 5.8S, and 25S rRNAs requires removing the “spacer” sequences. The canonical pathway for the removal of the ITS1 spacer involves cleavages at the 3′ end of 18S rRNA and at two sites inside ITS1. The process can generate either a long or a short 5.8S rRNA that differs in the number of ITS1 nucleotides retained at the 5.8S 5′ end. Here we document a novel pathway to the long 5.8S, which bypasses cleavage within ITS1. Instead, the entire ITS1 is degraded from its 5′ end by exonuclease Xrn1. Mutations in RNase MRP increase the accumulation of long relative to short 5.8S rRNA. Traditionally this is attributed to a decreased rate of RNase MRP cleavage at its target in ITS1, called A3. However, results from this work show that the MRP-induced switch between long and short 5.8S rRNA formation occurs even when the A3 site is deleted. Based on this and our published data, we propose that the link between RNase MRP and 5.8S 5′ end formation involves RNase MRP cleavage at unknown sites elsewhere in pre-rRNA or in RNA molecules other than pre-rRNA.

## 1. Introduction

Ribosome formation is the most resource-requiring process in both pro- and eukaryotes [1,2,3]. It involves complex pathways for regulated synthesis of rRNA and ribosomal proteins (r-proteins), and the assembly of these components into functional ribosomal subunits. The pathways differ between pro- and eukaryotes, but in eukaryotes the major steps are conserved from yeast to humans, although the complexity has evolved [4,5,6].

The progression of eukaryotic ribosome formation has been most intensively studied in *Saccharomyces cerevisiae* (“yeast”) [7,8,9]. In this work, we focus on the processing of the yeast rRNA. Three of the four eukaryotic rRNA molecules are generated from a single precursor rRNA (pre-rRNA), polymerized by Pol I, which contains the sequences for 18S, 5.8S, and 25S rRNA separated by the internal transcribed spacers ITS1 and ITS2, and flanked by 5′ and 3′ external transcribed spacers (ETS) (Figure 1A). The pre-rRNA is processed by endo- and exonucleases to form the mature rRNAs (Figure 1B) concurrently with the assembly of the rRNA and 79 r-proteins into the two ribosomal subunits [9]. The rRNA processing can take place during transcription or after completed synthesis of the pre-rRNA transcript [10,11].

The canonical model for processing of the primary Pol I transcript begins with the Utp24 and Rnt1 endonucleases splitting the 5′ ETS and 3′ ETS, respectively, from the main portion of the pre-rRNA, generating the 32S intermediate (Figure 1B) [12,13,14]. Utp24 then cuts at the A2 site in ITS1, which separates the rRNA moieties destined for the 40S or the 60S ribosomal subunit [12,14,15]. ITS1 is further processed to generate the 5′ end of the 5.8S rRNA and the 3′ end of 18S rRNA (see below). The 3′ end of 5.8S rRNA is generated by Las1 cleavage at the C2 site in ITS2 followed exonuclease cut-back by the exosome [16]; the downstream part of ITS2 is removed by exonucleases Rat1 and Xrn1 [17].

**Figure 1 ijms-22-06690-f001:**
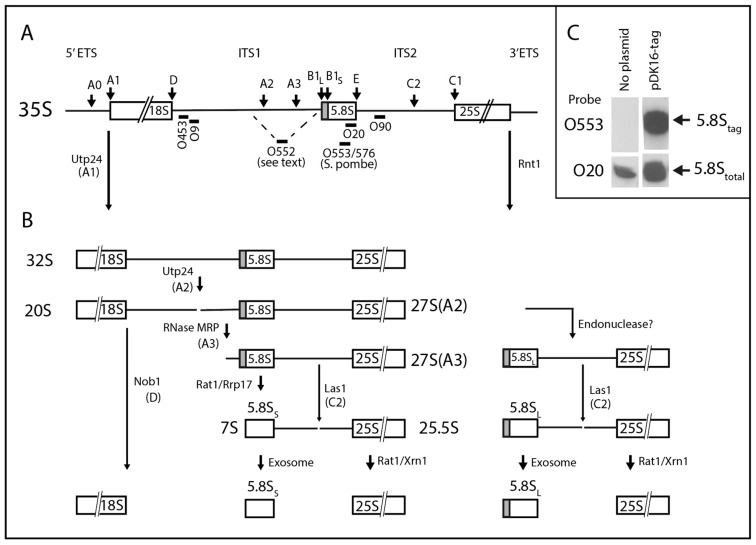
Canonical rRNA processing pathways. (**A**) Map of the yeast RNA polymerase I-transcribed rRNA transcription unit with processing sites (above map) and oligonucleotides used for probing northern blots (below map). See Table 1 for oligonucleotide sequences and positions of pre-rRNA to which they hybridize. (**B**) rRNA processing intermediates with names of relevant processing enzymes and their sites of action. Only relevant steps are shown. (**C**) Specificity of northern probe for tagged 5.8S rRNA. ∆*rpa12* without plasmid or carrying pDK16-tag wildtype was grown at 25° and shifted to 37° for 6 h. Total RNA was analyzed by northern analysis using the O553 probe (complementary to the *S. pombe* tag in 5.8S rRNA on pDK16-tag) or the O20 probe (complementary to the 25 nucleotides at the 3′ end of 5.8S rRNA).

There are two pathways that generate the 5′ end of the 5.8S. In the major pathway, the ribozyme RNase MRP cuts ITS1 at the A3 site, followed by trimming of the resulting 5′ end by a combination of Rat1 and Rrp17 exonucleases to form the 5′ end of the “short” 5.8S rRNA (5.8S_S_) (Figure 1 and Figure 2) [18,19,20,21,22,23,24]. The other pathway ostensibly involves an unknown endonuclease that directly generates the 5′ end of the “long” 5.8S rRNA (5.8S_L_), which is 7 nucleotides longer at the 5′ end than 5.8S_S_ [25]. Mutations in the RNA or protein subunits of RNase MRP favor the accumulation of 5.8S_L_, increasing the fraction of the total 5.8S rRNA constituted by 5.8S_L_ (“L-fraction”) [23,26,27,28,29,30,31]. It is generally believed that this is caused by a decreased rate of RNase MRP cleavage at A3, which in turn reduces the number of A3 5′ ends available for attack by Rat1 and Rrp17.

Here, we made two discoveries that implicate changes to the current model for 5.8S processing. First, processing of pre-rRNA carrying a deletion that removes both A2 and A3 sites was degraded by the exonuclease Xrn1 beginning from the 5′ end generated by D-cleavage and ending at the 5′ end of 5.8S_L_. This extended our previous observation that after inactivation of RNase MRP, processing of wildtype rRNA exclusively generates 5.8S_L_ and that this process also requires Xrn1 [30]. Hence, we proposed that the Xrn1-dependent pathway to 5.8S_L_ also occurred during processing of wildtype pre-rRNA. Second, we found that the L-fraction increases in an RNase MRP mutant, even if the A3 site in the pre-rRNA is deleted, indicating that the RNase MRP effect on the L-fraction is not the result of decreased A3 cleavage. Consequently, we suggest that RNase MRP modulation of the level of 5.8S_L_ is indirect.

## 2. Results

### 2.1. System for Genetic Analysis

The *S. cerevisiae* genome has 100–200 tandem copies of the 18S-5.8S-25S rRNA transcription unit that is transcribed by RNA Pol I [32]. The large number of rRNA genes makes traditional genetic manipulation difficult and we therefore used strains in which Pol I is inactivated either permanently (∆*rpa135*) or conditionally at 37° (∆*rpa12*) [33,34] (Table 2). Both strains harbor an 18S-5.8S-25S rRNA transcription unit on the high-copy plasmid pDK16 that is transcribed by RNA polymerase II (Pol II) from the Cu^2+^ induced CUP1 promoter [35]. Previous studies showed that pre-rRNA transcripts produced by Pol II are processed through the same steps as the natural Pol I transcripts [33,35]. 

To discriminate between the 5.8S rRNA transcribed from the plasmid-borne Pol II-driven rRNA genes and the chromosomal Pol I rRNA genes, we tagged the 5.8S rRNA derived from pDK16 by replacing the DNA sequence corresponding to a hairpin formed by nucleotides 124–147 of the 5.8S rRNA with the DNA sequence that generates the same hairpin in *Schizosaccharomyces pombe* (Sp) 5.8S rRNA, albeit with a different nucleotide sequence. The modified 5.8S rRNA transcribed from this plasmid, called pDK16-tag, was detected on northern blots by probing with an oligonucleotide (O553 or O576) that is complementary to the *S. pombe* sequence (Table 1). Total 5.8S rRNA transcribed from both wildtype and tagged genes was visualized on northern blots by a probe, O20, which is complementary to the 25 nucleotides at the 3′ end of 5.8S rRNA and thus present in both wildtype and Sp-tagged 5.8S rRNA (Figure 1C). 

### 2.2. Effects of ITS1 Deletions on 5.8S rRNA Processing

Previous experiments have shown that the canonical A2 and A3 cleavage sites in ITS1 are dispensable for formation of mature rRNA [21,35]. To determine if other regions in ITS1 are necessary for 5.8S rRNA formation, we generated several ITS1 deletions in the rRNA transcript on pDK16-tag (Figure 2A). The deletion plasmids were transformed into the temperature-sensitive strain ∆*rpa12* to test if the ITS1-rRNA deletion genes can support growth at the non-permissive temperature for RNA Pol I in the presence of Cu^2+^, the inducer of the *CUP1* promoter driving transcription of the plasmid-borne rRNA transcription unit (Appendix A, see Materials and Methods for specifics). Somewhat surprisingly, only one deletion (∆4), removing 82 nucleotides in the upstream part of ITS1, was lethal at 37°, and a shorter deletion within the same region of ITS1 (∆7) resulted in severely impeded growth (Appendix A). All the remaining deletion plasmids supported growth at 37° (Appendix A and not shown). To confirm the ability of the longest ITS1 deletion (∆2) to provide functional rRNA, we transformed it into the ∆*rpa135* mutant and found that it supports growth in the presence, but not in the absence, of Cu^2+^ (Appendix A, see Materials and Methods for specifics).

We next asked if the viable deletions affect the relative accumulation of the long (5.8S_L_) and short (5.8S_S_) rRNAs. For these experiments, we chose a derivative of ∆*rpa12* in which we had deleted the *XRN1* gene, because internal ITS1 fragments are stabilized in the absence of the Xrn1 exonuclease [36]. Cultures of this host carrying pDK16-tag or its deletion derivatives were grown at 25° and shifted to 37° for 6 h. Total RNA was analyzed by acrylamide gel electrophoresis and northern blot hybridization, using the oligonucleotide probe O553 (complementary to the 5.8S Sp tag). As seen in Figure 3 the fraction of 5.8S rRNA constituted by the long form (L-fraction) increased very noticeably when rRNA was processed from several of the mutant plasmids. Moreover, bands of relatively low mobility appeared in some lanes, but there is not a one-to-one correlation between the increased L-fraction and the appearance of the slow-moving bands (Appendix A). To determine if the 5′ ends of the two types of 5.8S match those resulting from processing of wildtype pre-rRNA, we expressed pDK16-tag with and without the ∆2 deletion in ∆*rpa135* and used total RNA for primer extension of O20 that is complementary to the 3′ end of the mature 5.8S (Figure 4). The results show that positions of the 5′ ends of both the short and long 5.8S rRNA match the 5′ ends observed when rRNA is expressed from wildtype chromosomal genes, i.e., the 5′ end of 5.8S_L_ retains 7 nucleotides more from the ITS than does 5.8S_S_ [31,37].

We then calculated the ratio between the long (5.8S_L_) and short (5.8S_S_) 5.8S rRNA (L-fraction) produced from each of the deletion pre-rRNAs. Since there was no difference between the results between rRNA extracted from cells growing at 25° or shifted to 37° (Appendix A), we merged results from the two sets of samples. (It was expected that the L-fraction would be very similar at the two temperatures, even though the rRNA processing pattern is changed by 4–6 h after the temperature shift due to inactivation of the ∆*rpa12* PolI (see below). This is because processing intermediates turn over relatively quickly (in wildtype the processing of pre-rRNA takes about 10 min), while the pool of the stable 5.8S end products only changes with cell growth.).

For wildtype ITS1, the L-fraction was 0.21, i.e., the cells accumulated about 4-fold more 5.8S_S_ than 5.8S_L_, consistent with the L-fraction in 5.8S rRNA transcribed from chromosomal genes (Figure 5A; Appendix A) [31,39]. Deletions removing the A2 site (∆13), the A2 and A3 sites together (∆14), the A3 site by itself (∆15), or a proposed hairpin upstream of the A3 site (∆16) all have no or modest (≤1.5-fold) effects on the L-fraction (Figure 5A; Appendix A). However, a 10-nucleotide deletion immediately downstream of the A3 cleavage site (∆18) increased the L-fraction more than 2-fold. Importantly, a deletion that leaves the A2 and A3 sites intact but removes 41 nucleotides downstream of the A3 region (∆17) increases the L-fraction about 3-fold, showing that deletions in the distal end of ITS1 have stronger effects than the loss of the A3 site. This idea is supported by the increased L-fraction in ∆2 and ∆19, both of which lack portions of the region downstream of A3. Although these deletions also remove one or both of the A2 and A3 sites, comparison with the other deletions lacking the nucleolytic sites rule out that the presence or absence of A2 and A3 is of major importance for the balance between the short and long 5.8S rRNA. The salient features of the ITS1 deletions expressed in ∆*rpa12* ∆*xrn1* and other strains used for the following experiments are summarized in Table 3. 

### 2.3. pre-rRNA Processing Is Altered by an RNase MRP Mutation Even When the ITS1 A3 Site Is Deleted

The 5.8S L-fraction increases in RNase MRP mutants [22,23,26,27,28,29,30,31]. This increase has been attributed to a reduced rate of A3 cleavage with an ensuing decrease in the number of A3 5′ ends available for attack by Rat1/Rrp17 to generate 5.8S_S_. If this explanation is true, then 5.8S rRNA formation in mutants lacking the A3 site should not be affected by RNase MRP mutations. To examine this, we transformed plasmid pDK16-tag and its deletion derivatives into two isogenic strains that either contain the wildtype gene for the RNase MRP RNA subunit (*RRP2* alias *NME1*) or its temperature-sensitive sibling, *rrp2-2*, with a single-base substitution in the enzyme’s RNA component (Table 2) [23]. Both of these strains have a wildtype XRN1 gene. Note, however, that even though the mutant grows at 25° and not at 37°, there is little difference in the relative abundance of long and short 5.8S rRNA at the two temperatures [31]. 

Total RNA from these strains was analyzed by northern blots for 5.8S_S_ and 5.8S_L_ derived from the Sp-tagged plasmids (Figure 3 middle and bottom panel). The L-fractions for RNA derived from the ITS1 deletion genes in the *RRP2* strain differed little from the corresponding data measure in the ∆*rpa12* ∆*xrn1* background (Figure 5A; Appendix A). An average of the results in the two strains are given in Figure 5B and Appendix A. However, the low-mobility bands seen in ∆*rpa12* ∆*xrn1* were not seen in the *RRP2* strain, suggesting that they are due to the lack of Xrn1 activity (Appendix A).

As expected, the L-fraction of 5.8S rRNA synthesized from the pDK16-tag plasmid without an ITS1 deletion is increased in the *rrp2-2* mutant relative to the *RRP2* parent strain (Figure 3, compare lanes 23–24 and 35–36 with lanes 45–46 and 57–58, respectively; quantified in Figure 5 and Appendix A). The L-fraction is also increased in *rrp2-2* harboring pDK16-tag ∆13 and ∆16 (Figure 3, compare lanes 27–28 and 37–38 with lanes 49–50 and 59–60, respectively); both of these deletions leave the A3 site intact. Remarkably, the *rrp2-2* mutation also changes the relative amount of 5.8S_L_ in mutants ∆14 and ∆15 (Figure 3, compare lanes 29–30 and 31–32 with lanes 51–52 and 53–54, respectively; quantified in Figure 5 and Appendix A), even though A3 is deleted in these mutants. In contrast, the RNase MRP mutation has little effect on the L-fraction in mutants ∆2, ∆17 and ∆18, in which the L-fraction is already increased in the *RRP2* wildtype background due to the deletions (Figure 5, compare lanes 25–26, 39–40, and 41–42 with lanes 47–48, 61–62, and 63–64, respectively).

### 2.4. A Pathway to 5.8S rRNA That Bypasses the Canonical A2 and A3 Cleavage Sites in ITS1

The accumulation of 5.8S rRNA resulting from processing of the ∆2 and ∆14 mutants (Figure 2, Figure 3, Figure 4 and Figure 5) shows that the canonical path to 5.8S rRNA via cleavage at A2 and/or A3 can be bypassed. To learn more about the bypass pathway(s), we first asked if the mutant pre-rRNAs are cleaved accurately at the D site at the 3′ end of mature 18S rRNA. Total RNA from ∆*rpa12* cells harboring the pDK16-tag plasmid or the ∆2, ∆13, ∆14, and ∆15 deletion derivatives was prepared 6 h after a shift from 25° to 37°, at which time wildtype processing intermediates derived from residual transcription of the chromosomal rRNA genes are no longer visible (see below). Total RNA was used as template for extension of oligonucleotide O9, which hybridizes to ITS1 24–53 nucleotides downstream of the D site (Figure 1A and Table 1). RNA from the wildtype and the ∆2, ∆13, and ∆14 strains generated bands matching the 5′ end of ITS1, adjacent to the 3′ end of mature 18S rRNA (Figure 6, lanes 2–5). Since both the A2 and A3 sites are deleted from ∆2 and ∆14, this demonstrates that accurate D cleavage does not depend on prior cleavage at A2 or A3. Interestingly, in mutant ∆15 the band corresponding to cleavage at D is absent; instead, there is a band corresponding to ectopic cleavage about 10 nucleotides downstream of the proper D site (Figure 6, lane 6, “D-~10”). A secondary band at “D-~10” is also seen in ∆13 (Figure 6, lane 4). It is interesting to note that the in vitro incubation of purified Nob1, the enzyme that cleaves the D-site (Figure 1B), with a short RNA substrate modeling 24 nucleotides around the 18S-ITS1 boundary region, also results in cutting at the precise D-site with a secondary target site about 10 nucleotides downstream of the D-site [40], supporting the idea that the D-site target of Nob1 is defined in the local region around the D-site. 

There is an additional interesting result from the primer extension experiment: RNA from ∆2 (and, to a lesser extent, ∆13 and ∆14) generates a longer band whose length corresponds to the position of adenosines 1781–1782 near the end of 18S rRNA that are dimethylated in the mature rRNA [41] (Figure 6, lanes 3–5). This band is not visible with the wildtype RNA, presumably because the D-cleavage normally occurs at a faster rate, thereby preceding the A-methylation. We concludethat in the ∆2, ∆13, and ∆14 mutants, the D site cleavage is delayed relative to A-dimethylation, and that 3′ end maturation of 18S rRNA is not a prerequisite for dimethylation. Similar results were seen previously in A2 site mutants [21,35].

To gain insight into the role of the Xrn1 exonuclease in the processing of the ∆2 pre-rRNA, we compared northern blots of RNA from ∆*rpa12* with and without the *XRN1* deletion, which stabilizes ITS1 processing fragments [38]. RNA was prepared from ∆*rpa12* ∆*xrn1* at 4 and 6 h after a shift from 25° to 37° and analyzed on northern blots of agarose and acrylamide gels probed with O453 (Figure 7A), which is complementary to nucleotides 15–25 of ITS1 (Figure 1A and Table 1). As previously observed [38], the elimination of Xrn1 activity with the wildtype ITS1 plasmid results in the accumulation of an RNA generated from cleavage at sites D and A2 (D-A2) (Figure 7A, lanes 1–2, and lanes 7–8). At 4 h post-temperature shift, the ∆2 mutant contained accumulated D-A2 RNA; we attribute that to residual but decreased transcription of the chromosomal rRNA genes by Pol I as it was undergoing inactivation after the temperature shift. However, the blot also revealed a longer fragment, labeled X, derived from the processing of ITS1 from mutant ∆2 (Figure 7A, lanes 3–4 and lanes 9–10). By 6 h, the D-A2 fragment was virtually gone, and only band X was visible (Figure 7A, lanes 5–6 and lanes 11–12). Furthermore, the wildtype 20S (pre-18S) band generated by cleavage at A2 (Figure 1; barely visible in Figure 7A, lanes 1–2) was replaced in the ∆2 mutant with a new “24S” slower moving band (Figure 7A, lanes 3–6). This RNA may be the same fragment we have previously observed in an RNase MRP mutant, which extends from the 5′ end of 18S to the 3′ end (E site) of 5.8S rRNA [23,31]. Finally, several bands between X and 20S were observed after the temperature shift of ∆*rpa12* ∆*xrn1*/pDK∆2, indicating ectopic cleavage of 18S rRNA (Figure 7A, compare lanes 3–6 with lanes 1–2). 

Given the size of the ∆2 deletion, that ∆2 pre-rRNA is cleaved at the proper D site, that O453 hybridizes to ITS1 positions 15–25, and the fact that band X is longer than D-A2, the X fragment must have a 5′ end at or very close to the D site and extend downstream of ITS1. We confirmed that X includes 5.8S by probing a blot of an acrylamide gel with the 5.8S-specific probe O20, which is complementary to the 3′ 25 nucleotides of 5.8S. Probing with O20 of the blot of RNA from ∆*rpa12* ∆*xrn1*/pDK16-∆2-tag revealed two fragments, one of which corresponds to X (Figure 7B, lane 5). The other, called X’, migrates slightly faster than X, but since it was not seen in the northern probed by O453, it must have a 5′ end downstream of the D site (Figure 7B, lane 5). Neither of the X and X’ bands hybridize to the O90 probe, complementary to the upstream part of ITS2 (Figure 7B, lane 11), suggesting that both X and X’ have 3′ ends at or close to the 3′ end of 5.8S rRNA (site E). Probing of the blot with O90 or O20 also reveals a band in the ∆*rpa12 XRN1*/pDK16-tag or the ∆*rpa12 XRN1*/pDK16-tag∆2 RNA corresponding to the 7S pre-rRNA (Figure 1A), which is the 3′ extended precursor for 5.8S rRNA bracketed by the B1 and C2 sites (Figure 7B lanes 1–3 and 7–9). This band is also seen in ∆*rpa12* ∆*xrn1*/pDK16-tag (Figure 7B, lanes 4 and 10). However, since the 7S band is not seen in ∆*rpa12* ∆*xrn1*/pDK16-tag∆2 probed with O90, we concluded that, in the absence of Xrn1 activity, the 3′ ends of X and X’ must be formed before the 5′ end of 5.8S is processed (Figure 7B, lane 11). 

The probing of RNA from ∆*rpa12* ∆*xrn1*/pDK16-tag∆2 with O20 also showed a series of bands below X and X’ (Figure 7B, lane 5), suggesting that the X and X’ RNA fragments are gradually shortened by exonuclease processing. To investigate this further, we designed a probe (O552) specific to RNA intermediates that included the ∆2 deletion (Figure 1A). This oligonucleotide is complementary to 9 nucleotides upstream and 11 nucleotides downstream of the ∆2 deletion in ITS1. Transcripts that encompass the ∆2 deletion can form a stable 20-basepair uninterrupted helix with the probe, while transcripts that do not contain the deletion are only able to form a 9- or 11-bp uninterrupted helix with O552, or a hybrid interrupted by a large loop, neither of which is stable enough to generate a hybrid under our hybridization conditions. As seen in Figure 7C, lanes 3 and 5, the O552 oligonucleotide only hybridizes to RNA from strains harboring pDK16-tag∆2, confirming that O552 specifically reveals transcripts that span the ∆2 deletion. Moreover, RNA from both the ∆*xrn1* and *XRN1* strains containing pDK16∆2 form exonuclease degradation “ladders” with O552 (Figure 7C, lanes 3 and 5, respectively), but the intensity of ladders in RNA from ∆*xrn1* are much greater (compare lane 3 with lane 5). The blot in Figure 7C, lanes 1–5, was stripped and then hybridized consecutively with O553, specific to the Sp-tag (Figure 7C, lane 6–10), and finally with O20, which hybridizes to both tagged and untagged 5.8S rRNA (Figure 7C right bottom). The results show that there were similar amounts of tagged 5.8S rRNA in both samples (Figure 7C, lanes 8 and 10). Therefore, we ascribe the increased intensity in lane 3 relative to lane 5 to a slower degradation rate of ITS1 fragments when the Xrn1 endonuclease is absent, leading to a greater accumulation of the intermediates [42]. We conclude that Xrn1 is the primary exonuclease generating the ladders and that, in the absence of the Xrn1 nuclease, the degradation is performed by another, slower 5′ > 3′ exonuclease.

Comparison of lanes 3 and 5 with lane 8 in Figure 7C shows that the bands at the top of the lower cluster of exonuclease products have an electrophoretic mobility similar to the X and X’ bands, while the lower bands migrate almost as fast as 5.8S_L_. These results suggest that the lower ladders represent “trimming” of X and X’ RNA to form the mature 5′ end of 5.8S (Figure 7C, lanes 3 and 5). The upper ladder may represent trimming of RNA formed by misprocessing of 18S rRNA. 

### 2.5. 5.8S rRNA with 5′ Extended Ends Are Incorporated into Functional Ribosomes

Since ∆2 rRNA supports growth at the non-permissive temperature of strain ∆*rpa12*, when the pDK16-tag∆2 plasmid is the only source of rRNA, functional ribosomes must be formed from the pre-rRNA containing the ∆2 deletion (Appendix A). We were curious to see if any of the 5′ extended 5.8S rRNA in X, X’, and the exonuclease ladders are removed before or after the 5.8S rRNA is incorporated into ribosomes. Therefore, we fractionated by sucrose gradients whole-cell extracts of ∆*rpa12* ∆*xrn1*/pDK16-tag∆2 (Figure 8). RNA extracted from each sucrose gradient fraction was analyzed on a northern blot probed with O576, which visualized the tagged 5.8S rRNA (Figure 8A). To locate the relevant ribosome peaks in the gradient, we quantified the relative amount of tagged 5.8S rRNA in each fraction (Figure 8B). The results demonstrate that the 5.8S_L_ processed from the ∆2 pre-rRNA is incorporated into 60S and 80S ribosomes as well as polysomes (Figure 8A, lanes 14–19 and lanes 21–27, respectively). Additionally, probing with O552 showed that the 5′-extended 5.8S RNA transcripts (“exoladders”) were also incorporated into ribosomes, including polysomes (Figure 8A, lanes 42–55). In other words, the extensions did not preclude incorporation of the 5.8S-containing rRNA into functional ribosomes. The distribution of transcripts of different lengths in the exonuclease ladders was somewhat different in the transcripts extracted from ribosomes compared to the total RNA extracted in the presence of hot phenol (Figure 8A, lanes 29 and 56, see also Figure 7C, lanes 3 and 5), presumably because hot phenol used for RNA extraction from whole cells inactivates enzymatic activity immediately upon cell lysis, while some enzymatic activities, including exonucleases, may continue to operate in the crude lysates despite keeping the lysates on ice. Interestingly, “24S” rRNA containing the ∆2 mutant ITS1 is seen in complexes that sediment relatively slowly (Figure 8A, lanes 4–8), but not in mature ribosomes. Thus, it appears that the “24S” transcript may be incorporated into slowly-sedimenting assembly intermediates (i.e., assembly intermediates in the early part of the assembly pathway) and that the 18S part is separated from the ITS1–5.8S part before ribosomes become functional. Finally, we note that the L-fraction (5.8S_L_/(5.8S_L_ + 5.8S_S_) is constant across the 60S, 80S, and polysome peaks, showing that neither of the two 5.8S forms are discriminated against in the 60S assembly process.

## 3. Discussion

### 3.1. Steps in the Xrn1-Dependent Path to the 5′ End of 5.8S_L_

The canonical scheme for pre-rRNA processing includes pathways to generate two different 5′ ends, 7 nucleotides apart, of the 5.8S rRNA (Figure 1A). The 5′ end of 5.8S_S_ (short 5.8S) is ostensibly created by cleavage at the A3 site in ITS1 by the endonuclease RNase MRP, followed by trimming by the exonucleases Rat1 and Rrp17 (Figure 1A and Figure 2) [19,20,21,22,23,24]. The 5′ end of 5.8S_L_ (long 5.8S) is believed to be formed by an unknown endonuclease [25] (Figure 1A). However, a third pathway for 5.8S processing was implied by our previous observation that formation of 5.8S_L_ requires exonuclease Xrn1 at non-permissive temperature in a temperature-sensitive RNase MRP mutant (“Mini2”). This mutation has a higher penetrance for RNase MRP inactivation than the *rrp2-2* mutant used in Figure 3 and Figure 5 [30]. 

In this report, we describe experiments that support the proposed Xrn1-dependent path to 5.8S_L_. By deleting most of the downstream-half of ITS1, including the A2 and A3 sites (∆2 mutant, Figure 2A,J), we blocked the two canonical processing pathways. This, in turn, uncovered a pathway, initiated by cleavage at the border between 18S and ITS1 (D site) followed by Xrn1 exonuclease trimming from D to the B1_L_ 5′ end (Figure 6, Figure 7, and Figure 9A). The removal of ITS1 sequences by Xrn1 occurs, at least in part, after the 5′ end-extended 5.8S rRNA is incorporated into functional 60S subunits, as indicated by the presence of ITS1 sequences from ∆2 pre-rRNA in the 60S-80S-polysome region of sucrose gradients (Figure 8). This observation is consistent with previous studies showing that large ribosomal subunits in both bacteria and yeast can be functional even if the rRNA is incompletely processed [43,44].

This new pathway shares two characteristics of the 5.8S_L_ formation from wildtype rRNA genes after temperature inactivation of RNase MRP in the *rrp2-*Mini2 mutant [30]. First, in neither case is ITS1 cleaved at either A2 or A3, and second, Xrn1 is implicated in both cases (see Figures 5 and 7 in [30]). These shared features suggest that, in the absence of RNase MRP activity, processing of wildtype pre-rRNA follows the pathway for ∆2 pre-rRNA processing. Moreover, northern analysis of RNA from the Mini2 mutant prior to RNase MRP temperature inactivation reveals an exonuclease ladder above 5.8S_L_ rRNA (see Figure 7 in [30]). This suggests that Xrn1 trimming is a normal pathway to 5.8S_L_ even in the presence of active RNase MRP. We concluded that the 5′ end of 5.8S_L_ can be formed either by Xrn1 or by an endonuclease while RNase MRP is active, but only the Xrn1-dependent pathway functions when RNase MRP is inactive (Figure 9B,C). 

### 3.2. The RNase MRP-Induced Switch between 5.8S_L_ and 5.8S_S_ Production Not Due to Changes in A3 Cleavage

The canonical role of RNase MRP in 5.8S_S_ processing (Figure 1) is based on two observations. First, RNase MRP cleaves ITS1 rRNA at the A3 site in vitro [18,46,47], and second, accumulation of 5.8S_S_ is decreased in RNase MRP mutants, ostensibly because of a decreased rate of RNase MRP cleavage at A3. If that were the case, no 5.8S_S_ should be made from pre-rRNA lacking the A3 site. However, processing from ∆14 and ∆15 pre-rRNAs yield normal amounts 5.8S_S_ even through these mutants lack the A3 site. In contrast, the ∆17 mutant that does have an A3 site yields more 5.8S_L_ than 5.8S_S_. Moreover, the balance between 5.8S_S_ and 5.8S_L_ production is lower in the *rrp2-2* mutant than in the *RRP2* wildtype sibling despite the lack of an A3 site (Figure 3 and Figure 5). Together, these observations suggest that, in contrast to the canonical model, the RNase MRP-induced switch between production of 5.8S_S_ and 5.8S_L_ cannot depend on the rate of A3 cleavage. Rather, we suggest that RNase MRP mediates the switch between 5.8S_S_ and 5.8S_L_ by an indirect, rather than a direct, mechanism. The absence of A2 cleavage after inactivation of RNase MRP (Figure 5 in [30]) suggests that such a mechanism involves a blockade of A2 cleavage.

It should be noted that, even though 8 nucleotides around the A3 consensus sequence contact the RNase MRP binding pocket, only two nucleotides in the consensus sequence have a strong effect on the rate of cleavage [46,47,48]. Thus, we cannot exclude ectopic RNase MRP cleavage of ∆14, ∆15, and ∆16 in the *RRP2* strain, although if this were the case, we would have expected that such misplaced cleavage should also suppress the effect of the inhibition of A3 cleavage in the ∆18 mutant.

The RNA component of RNase MRP is essential for rRNA synthesis and growth in both yeast and humans [30,49]. However, the essential nature of RNase MRP RNA cannot be rationalized based on our understanding of its role in rRNA synthesis, since both 5.8S_S_ and 5.8S_L_ become incorporated into functional ribosomes (Figure 8). An investigation of the proposed alternate model for RNase MRP-dependent inactivation of A2 cleavage may shed some light of this matter

### 3.3. The 3′ End of ITS1 Facilitates the Xrn1-Mediated Processing

The ITS1 mutants giving rise to preferential formation of 5.8S_L_ over 5.8S_S_ also accumulate high-molecular fragments in the ∆*rpa12* ∆*xrn1* mutant, but not in the *RRP2* and *rrp2-2* strains, suggesting that the absence of Xrn1 causes a kinetic retardation of an early step in ITS1 processing (Table 3 and Appendix A). These mutants (∆2, ∆17, ∆18, and ∆19) also lack part or all of the distal ITS1 (nucleotides 290–340). The accumulation of the high-molecular weight fragments that the distal part of ITS1 facilitates the processing in the absence of Xrn1. We suggest that the distal ITS1 might facilitate folding of ITS1, or perhaps be the target for binding of rRNA processing factor(s).

### 3.4. The 3′ End Maturation of 18S rRNA

The 3′ end of 18S rRNA is matured by Nob1 endonuclease cleavage at the D-site of the 20S pre-rRNA [50]. The D-site cut is cut accurately in vivo ∆2, ∆13, and ∆14 (Figure 6) and in vitro in a substrate with just 24 nucleotides around the 18S-ITS1 border [40], indicating that the Bob1 target is defined locally. However, cleavage in ∆15, and partially in ∆13, at an alternate site shows that a region of ITS1 more than 200 nucleotides downstream of the D-site also influences the accuracy of Nob1 cleavage, perhaps due to loss or gain of specific secondary or tertiary structures of ITS1.

Nob1 binds, together with partner proteins Pno1 and Nop9, to both 18S and ITS1 sequences of the 20S pre-rRNA in the nuclear pre-40S particle; cleavage is then prevented until helicases remove the partner proteins [51,52,53]. The ∆4 and ∆7 deletions remove parts of the pre-rRNA binding sites for Nob1 and partner proteins, which may account for the negative effect on growth of those mutations. 

Because the D-A2 fragment can be found in the cytoplasm [54,55], D cleavage is assumed to occur after export of 40S ribosomal precursors particles from the nucleus to the cytoplasm. However, this does not exclude that D cleavage could also occur in the nucleus prior to export of the pre-40S, since Nob1 binds to pre-40S particles in the nucleus and its inhibitory proteins may conceivably be removed by nuclear helicases. Our results suggest that D cleavage does in fact occur in the nucleus. Since both known endonuclease targets within ITS1 (A2 and A3) have been deleted in ∆2, the recognized mechanism for separating the rRNA moieties destined for 40S and 60S ribosomes is blocked. Therefore, we propose that, in the ∆2 mutant, the rRNA for the two subunits is instead separated by D cleavage. However, since nuclear export of the large and small ribosomal subunits requires different export factors [56], it seems unlikely that 18S and 5.8S/25S would be exported together in a single pre-ribosome particle. Thus, we propose that D-cleavage is a nuclear function during processing of ∆2 pre-rRNA. The cellular location of D cleavage may be determined by kinetic competition between nuclear export of pre-40S and cleavage at D. If the export is fast, Nob1 cleavage occurs predominantly in the cytoplasm, but during slow pre-40S export, Nob1 cleavage may become a nuclear function. In the case of ∆2 pre-rRNA processing, export of pre-40S is likely blocked until the precursors for 40S and 60S have been separated. 

### 3.5. Diversity of rRNA Processing Pathways

Like the pathway for bypass of ITS1 A2 and A3 cleavage described here, other steps for ribosome formation can also be bypassed by suppressor mutations. For example, Nsa1, a participant in restructuring one of the nucleolar pre-60S particles, requires the Rix7 ATP-helicase in order to be released from the pre-60S particle, but this step can be circumvented by mutations in *EBP2* and *MAK5* that eliminate the need for Nsa1 in facilitating pre-60S restructuring [57]. Similarly, a mutational change in the multifunctional protein Rrp5 can bypass A2 cleavage [58]. Moreover, mutations in Rsr1, Rpf2, or uL5, proteins that form a ribosomal subparticle with the 5S rRNA before docking of 5S rRNA-uL5 in the pre-60S particle, can suppress the need for Cgr1 in the final positioning of 5S rRNA in the nascent pre-60S [59]. The theme for the bypass pathways may be that they co-exist in wildtype strains but are kinetically non-competitive. The suppressor mutations may change the kinetics of the bypass reaction relative to the canonical pathways, enabling these alternative pathways to become kinetically significant. 

## 4. Materials and Methods

### 4.1. Strains and Growth Conditions

Yeast strains used are shown in Table 2. In ∆*rpa135*, the gene for the largest RNA polymerase subunit is disrupted, which inactivates RNA Pol I at all temperatures [33]. The pNOY102 plasmid in ∆*rpa135* was replaced by pDK16 or pDK16∆2 by transformation and selection for TRP^+^ followed by counter-selection of pNOY102 (URA3) by growth on 5-fluoroorotic acid. 

In ∆*rpa12*, disruption of the gene for the smallest subunit of RNA-Pol I bestows temperature sensitivity for growth [34]. An *XRN1* deletion derivative of the temperature-sensitive strain ∆*rpa12* (called ∆*rpa12* ∆*xrn1*) was constructed by transforming ∆*rpa12* with a PCR fragment made from chromosomal DNA of an *xrn1* ∆Bgl1::*URA3* strain [60] and selecting for uracil prototrophy. Plasmids containing deletions within the long rRNA transcription unit were tested for their ability to support growth of transformants selected for tryptophan prototrophy. Transformants of ∆*rpa135* were tested for growth on glucose medium in the presence and absence of Cu^2+^ at 30°. Transformants of ∆*rpa12* were streaked on plates with or without Cu^2+^ and incubated at 25° and 37°. Examples of these tests are shown in Appendix A. Plasmids pDK23 (carrying a 7-base deletion in the ITS1 A2 site) and pDK16∆1 (lacking all DNA between ITS1 position 201 and 25S rRNA position 318) were used as positive and negative controls, respectively. Plasmids pDK16, pDK16∆2, and pDK23 supported growth of ∆*rpa135* on glucose medium containing Cu^2+^ and growth of ∆*rpa12* at 37° on glucose-Cu medium, while pDK16 containing the ∆1, ∆4, and ∆5 deletions do not. Growth of cells expressing only the ∆2 version of pre-rRNA shows that pDK16∆2 also supports ribosome formation. Similar tests of the remaining ITS1 deletion plasmids showed that pre-rRNA containing ITS1 deletions ∆13, ∆14, ∆15, ∆16, ∆17, ∆18, and ∆19 deletions also support growth (not shown).

Cultures of ∆*rpa135* were grown in supplemented synthetic glucose medium lacking tryptophan and including 0.1 mM Cu^2+^ [61] at 30°, while ∆*rpa12* and its ∆*xrn1* derivative were grown in the same medium at 25° in supplemented synthetic medium and shifted to 37° for 4 or 6 hours as indicated. 

### 4.2. Plasmids and Oligonucleotides 

Plasmid pDK16 [35], a yeast-*E.coli* shuttle plasmid containing both 2-Micron and *ColEl* origins of replication, harbors a wildtype copy of the yeast rRNA transcription unit controlled by the *CUP1* promoter. pNOY102 [33] carries a wildtype rRNA transcription unit expressed from the *GAL7* promoter. Deletions were made in pDK16 by joining together PCR fragments with anchors of restriction enzyme recognition sites made with pNOY102 as template.

Oligonucleotides used for northern and primer extension analyses are listed in Table 1. Two probes (O553 and O576) were made for the Sp-tag on pDK16. Both work for northern analysis, but O553 has a propensity for hairpin formation that appears to limit its capacity to work in primer extension.

### 4.3. Other Procedures

Total RNA for gel electrophoresis was extracted from 0.3 mL of culture (~10^7^ cells per ml) by mixing with buffer containing hot (~90 °C) phenol in TSE1 (0.02 M of Tris-HCl, pH 7.4; 0.2 M of NaCl; 0.04 M of Na-EDTA; 0.1% sodium dodecylsulfate) and immediately vortexed with glass beads followed by extraction with phenol-CHCl_3_-isoamyl alcohol (25:24:1) and then with CHCl_3_-isoamyl alcohol (24:1) [31]. Agarose gel electrophoresis was done on 1% agarose in TBE (90 mM Tris-borate, pH 8.3, and 2 mM EDTA-Na). Each lane was loaded with 3 µg total RNA (0.06 A260 units). Acrylamide gel electrophoresis was done on 8% gels in 0.5× TBE. Each lane was loaded with 5 µg of total RNA. RNA was transferred to nylon membranes by capillary blotting for agarose gels and electro-transfer for acrylamide gels. RNA was crosslinked to the membrane by UV irradiation and hybridized with ^32^P-end-labeled oligonucleotides, followed by exposing the blot to storage phosphorimager screens. Autoradiograms were scanned on an 860 Storm imager (Molecular Dynamics). Bands on the images were quantified using Adobe Photoshop version 22.4.1. Graphing and statistical analysis were done with Microsoft Excel for the Mac version 16.49. Whole cell extracts for sucrose gradient analysis were prepared from quick-chilled cells by vortexing with glass beads in gradient buffer (50 mM of Tris-acetate, pH 7, 50 mM of NH_4_Cl, 12 mM of MgCl_2_, and 1 mM of DTT) containing 50 μg cycloheximide per ml. Sucrose gradients (10–50%) were loaded with 20 A260 units and centrifuged at 40,000 rpm for 6 h at 4 °C using an SW40Ti Beckman rotor. Finally, fractions (500 μL) were collected [62]. Primer extension was performed by hybridizing with ^32^P-end-labeled O9 oligonucleotide followed by incubation with reverse transcriptase [63].

## Figures and Tables

**Figure 2 ijms-22-06690-f002:**
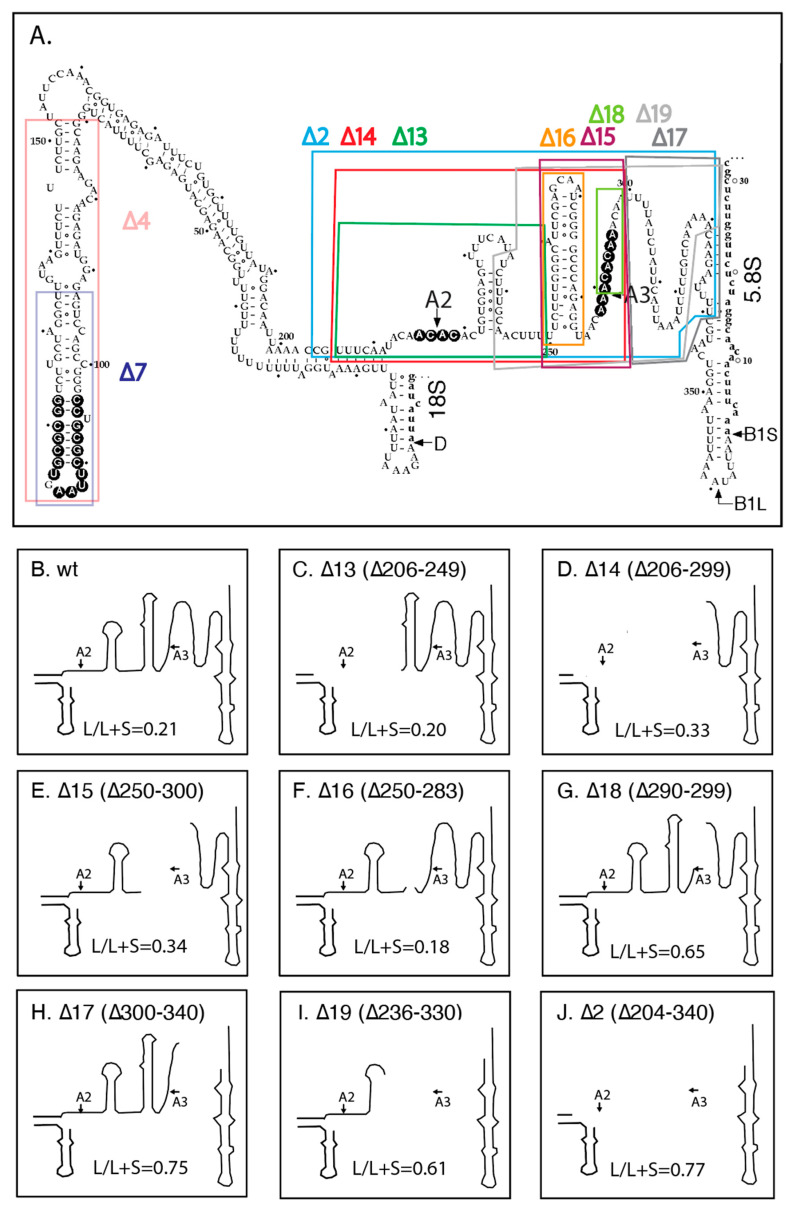
Deletion mutations in ITS1. (**A**) Secondary structure of ITS1 in *S. cerevisiae* proposed by van Nues at al. [38]. Deletions are shown by color-coded frames around the deleted nucleotides. Phylogenetically conserved nucleotides among a set of *Saccharomycetales* are shown in reverse contrast; for specifics see [36]. (**B**–**J**) Schematics of the ITS1 secondary structure in which the deleted parts in each mutant are blocked out. The total 5.8S rRNA constituted by 5.8S_L_ (“L-fraction”) was determined from northern blots loaded with RNA extracted from cells carrying each of the deletion plasmids. The numbers are the average of the measurements from 25° and 37° in ∆*rpa12* ∆*xrn1* and YLL53 (*RRP2*); the latter strain is not deleted for *XRN1*.

**Figure 3 ijms-22-06690-f003:**
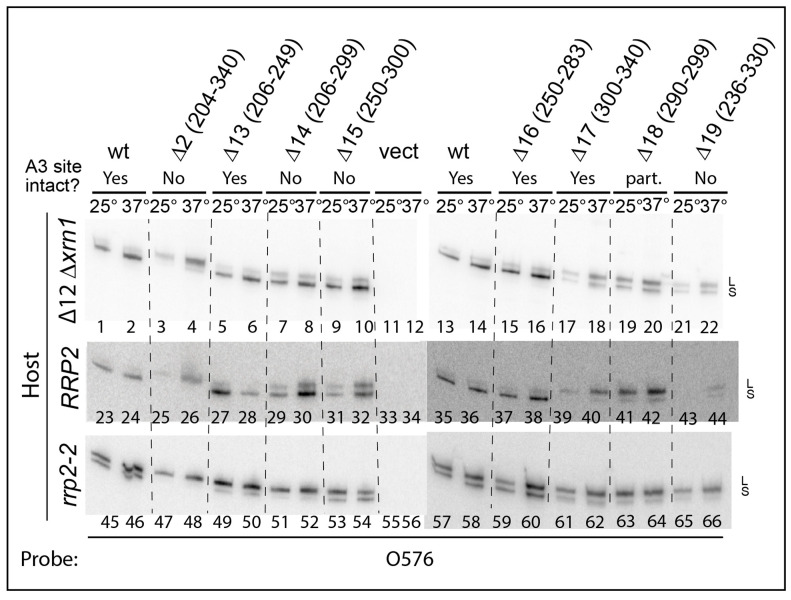
Northern analysis of RNA extracted from ∆*rpa12* ∆*xrn1* (top), *RRP2* (middle), and *rrp2-2* (bottom) each carrying pDK16-tag or one of its ITS1 deletion derivatives. Total rRNA was fractionated by electrophoresis on acrylamide gels and probed with O576 on northern blots. See Appendix A for uncropped autoradiograms.

**Figure 4 ijms-22-06690-f004:**
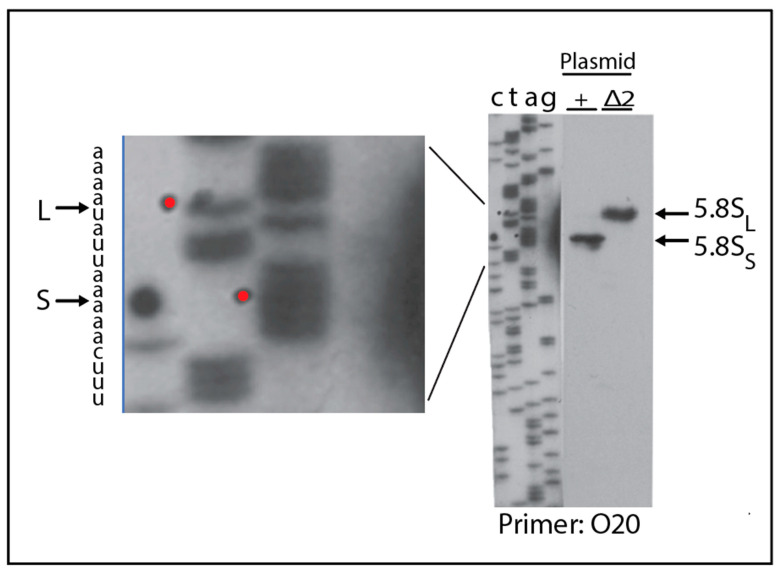
Mapping of 5′ ends of 5.8S rRNA processed from ∆2 and wildtype rRNA genes carried on plasmid pDK16. The host strain ∆*rpa135* carrying wildtype pDK16 or pDK16∆2 was grown in steady state at 30°. Total RNA was isolated and used for extension of primer O20. Di-deoxy sequencing ladders generated by extension of the O20 primer on the pDK16 DNA are shown as markers. Red dots in the sequence lanes indicate the bands that line up with the primer extension products. Note that the gel was loaded using a “shark-tooth comb”, meaning that there is minimal space between the slots. The primer extension products from the pDK16 and pDK16∆2 were each loaded twice in neighboring lanes, but since the lanes are close, the bands of the duplicate loadings virtually merged into a single double-width band. Furthermore, a few irrelevant lanes between the sequence ladder and the primer extension products were excised from the image.

**Figure 5 ijms-22-06690-f005:**
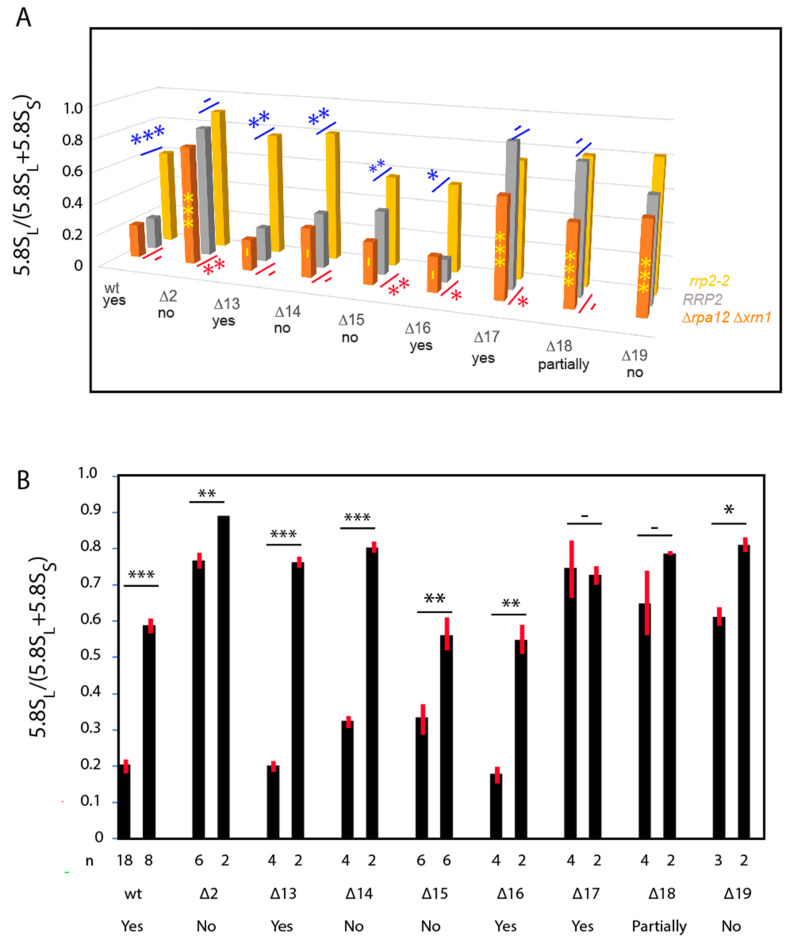
Bands in the northern blots shown in Figure 3 were quantified and the fraction of total 5.8S rRNA constituted by 5.8S_L_ was calculated. Data for 25° and 37° (Appendix A) were pooled for each strain (see text). (**A**) Data were plotted in a 3D format. Front row: ∆*rpa12* ∆*xrn1*; middle row: *RRP2* (YLL53); back row: *rrp2-2* (YLL54). The results for different strains were compared by a Student’s *t*-test. Yellow symbols: ∆*rpa12* ∆*xrn1* containing pDK16-tag deletion plasmids compared to ∆*rpa12* ∆*xrn1* with pDK16-tag (no ITS1 deletion); red symbols: comparing ∆*rpa12* ∆*xrn1* and *RRP2* harboring the same plasmid; blue symbols: comparing *RRP2* and *rrp2-2* harboring the same plasmid. (**B**) Results from ∆*rpa12* ∆*xrn1* and *RRP2* carrying a given plasmid were averaged (left bar for each plasmid). Results for *rrp2-2* are shown in the right bar for each plasmid. Standard deviations are shown by the red lines at the top of each bar (the standard deviation for ∆2 in the *rrp2-**2* strain was 0). Note that the *rrp2-2* strain grows at 25°, not at 37°, yet displays a similarly mutant RNA phenotype at both temperatures [31]. *** *p* < 10^−3^; ** *p* < 10^−2^; * *p* ≤ 5 × 10^−2^; - *p* > 5 × 10^−2^.

**Figure 6 ijms-22-06690-f006:**
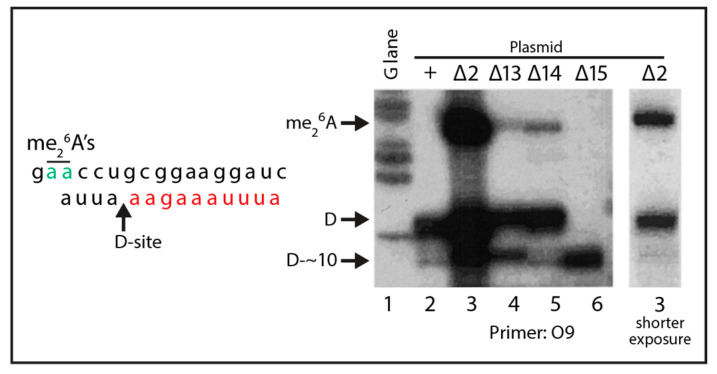
Mapping of 5′ ends at the D-site made by processing of pre-rRNA from wildtype and the indicated deletion mutants. Host ∆*rpa12* carrying the indicated derivatives of pDK16-tag was grown at 25° and shifted to 37° for 6 h. Total RNA was used for extension of primer O9 (Figure 1A). A G-lane made by dideoxy sequencing using pDK16 DNA and primer O9 is shown as markers. The blot on the right shows lane 3 from a shorter exposure of the blot in lanes 1–6. The nucleotide sequence around the D site is shown on the left to facilitate the interpretation of the G-lane as marker. Red: ITS1 nucleotides; black: 18S rRNA nucleotides; green: the dimethylated A residues.

**Figure 7 ijms-22-06690-f007:**
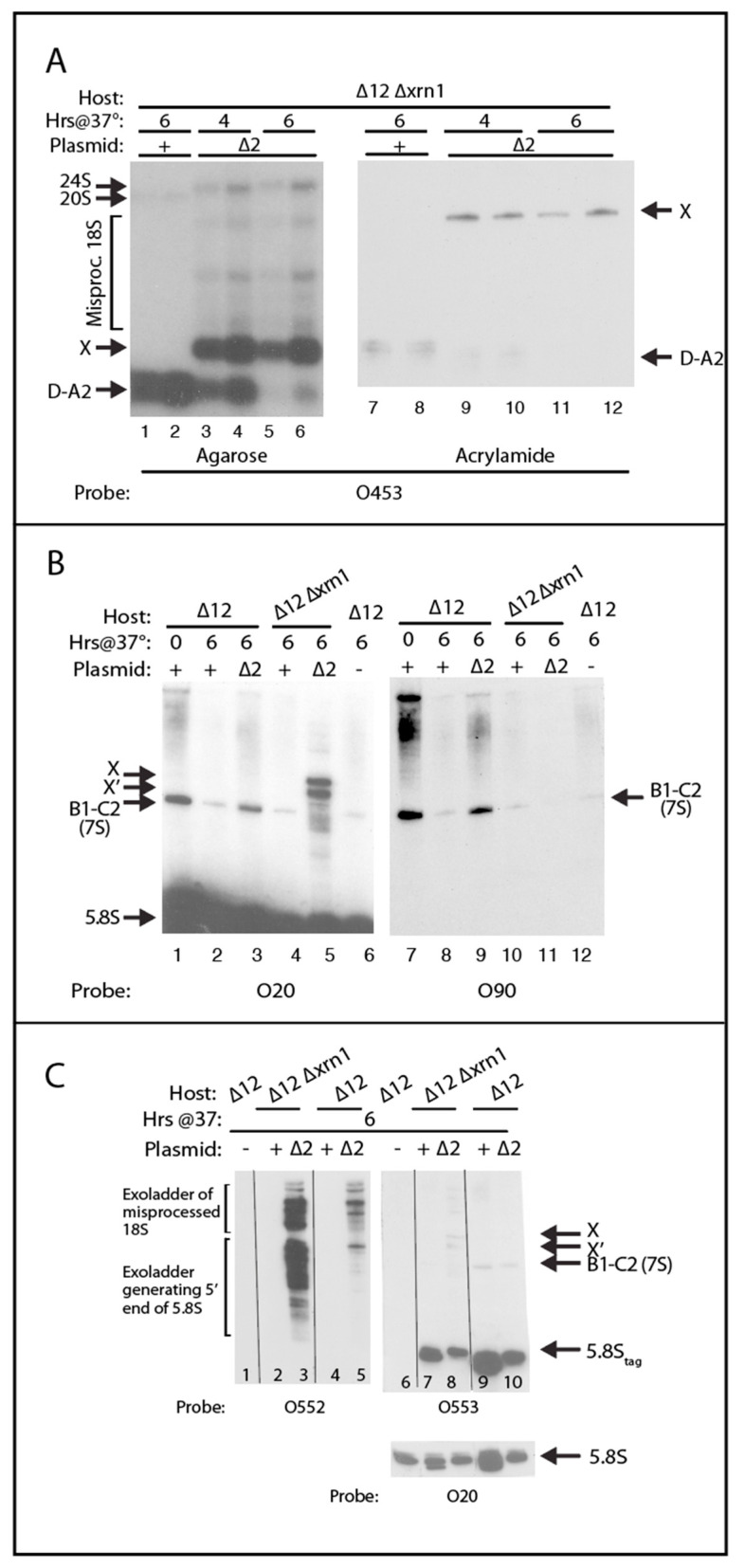
Characterization of intermediates in the processing of ∆2 pre-rRNA. Total RNA from ∆*rpa12 XRN1* or ∆*rpa12* ∆*xrn1* carrying pDK16-tag with wildtype or ∆2 rRNA genes was extracted from cells grown at 25° (time 0) or after cells were shifted to 37° for 4 or 6 h. The RNA was fractionated by agarose or acrylamide gel electrophoresis and analyzed by northern blots. (**A**) The wildtype D-A2 wildtype intermediate was replaced with a longer intermediate (X) in ∆2 processing. Agarose (left) and acrylamide gels (right) were blotted and probed with O453. (**B**) The X and X’ intermediates in processing of ∆2 pre-rRNA include the 3′ end of 5.8S rRNA. Acrylamide gels were blotted and probed with O20 or O90. (**C**) The ∆2 pre-rRNA processing intermediates were subject to exonuclease maturation. An acrylamide gel was blotted and probed with O552 (lanes 1–5), O553 (lanes 6–10), or O20 (5.8S slice of the blot showing the 5.8S bands in lanes 6–10). All images in panel (**C**) came from the same blot, probed sequentially and stripped in between each probing; see text for details. The blots in this figure were not cropped at the top or the bottom of the blot, except for the O20 loading control at the bottom right of the figure. Irrelevant lanes were cut out in panel (**C**) (indicated by the lines). The specificity of the O552 probe is also evident from the uncropped blot in Appendix A.

**Figure 8 ijms-22-06690-f008:**
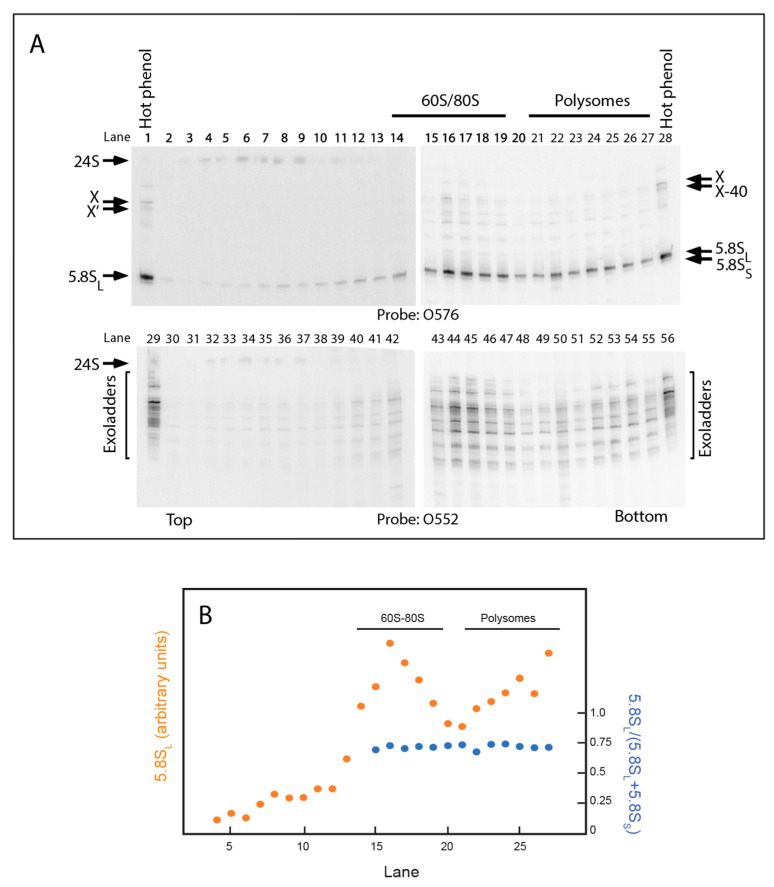
5′ extended 5.8S rRNA is incorporated into 60S, 80S and polysomal ribosomes. ∆*rpa12* ∆*xrn1* carrying pDK16-tag∆2 rRNA genes was grown at 25° and shifted to 37° for 6 h. Whole cell extracts were fractionated on sucrose gradients. RNA was then isolated from each fraction and subjected to acrylamide gel electrophoresis. Finally, RNA was transferred to nylon blots and probed with O576 or O552. (**A**) Northern blots. (**B**) Quantification of 5.8S_L_ (orange) and the L-fraction (5.8S_L_/(5.8S_L_ + 5.8S_S_) across the gradient.

**Figure 9 ijms-22-06690-f009:**
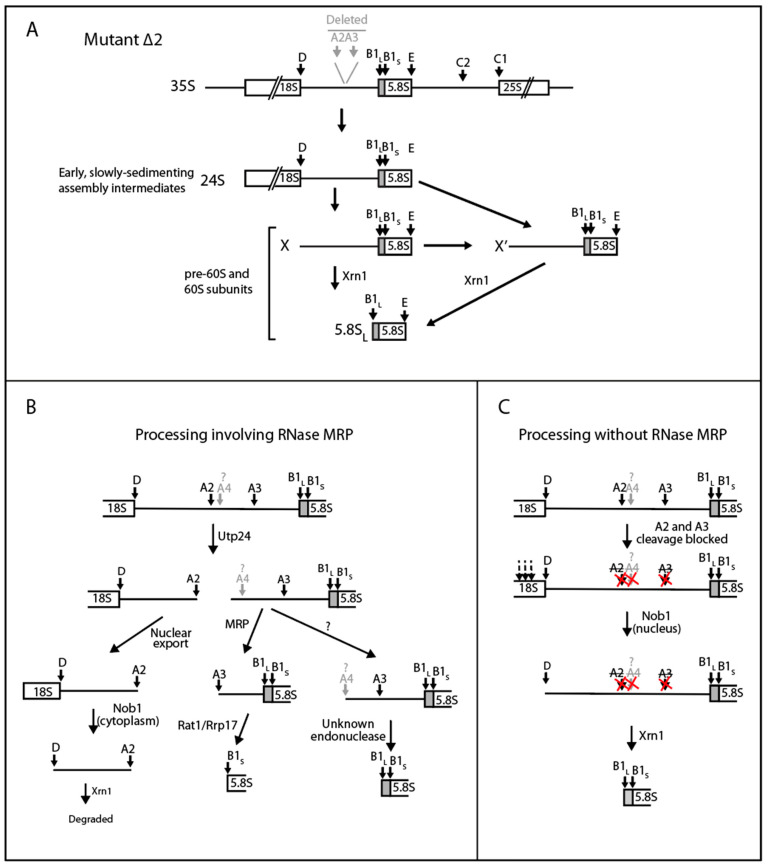
Models for ITS1 processing and switch between processing modes. (**A**) Xrn1-dependent processing of ∆2 pre-rRNA. See text for details. (**B**) Processing involving RNase MRP (canonical model). Note that the path to 5.8S_L_ involves an intermediate with a 5′ end (A4) a short distance downstream of A2 [25,45]. (**C**) Processing without RNase MRP participation. Inactivation of RNase MRP blocks cleavage at both A2 and A3, allowing Xrn1 exonuclease to degrade the entire region from D to B1_L_. We propose that the processing of pre-rRNA occurs via a competition between the pathways in panels B and C, with the canonical model in panel B being dominant during growth under laboratory conditions. If RNase MRP activity is decreased due to change of growth conditions or mutations, the kinetic mix of the models in panels B and C switches in the direction of model C. In the extreme case of total inactivation of RNase MRP, processing occurs exclusively via the model in C.

**Table 1 ijms-22-06690-t001:** Sequence of oligonucleotides used.

Oligo	Sequence (5’→3’)	Complementary to
O9	GCT CTT GCC AAA ACA AAA AAA TCC A	ITS1 24–53
O20	AAA TGA CGC TCA AAC AGG CAT GCC C	5.8S 3′ end
O90	GTA TCA CTC ACT ACC AAA CAG AAT G	ITS2 11–35
O453	AAC AAA AAA ATC CAT TTT CAA	ITS1 15–25
O552	CCA GTT ACG CGT TTT AAT TG	Spanning ∆2 deletion, see text
O553	ATG CCT TTG GTA GAA CCC AAA GGC	*S. pombe* hairpin inserted in *S. cerevisiae* 5.8S
O576	ATG CCT TTG GTA GAA CCC	*S. pombe* hairpin inserted in *S. cerevisiae* 5.8S

**Table 2 ijms-22-06690-t002:** Strains and plasmids.

Name	Genotype	References
∆*rpa12* alias NOY504	*MATa rm4 (rpa12)::LEU2 ade2-101 ura3-1 trp1-1 leu2-3,112 his3-101 can1-100*	[34]
∆*rpa135* alias NOY408-la	*MATa rpa135::LEU2 ade2-1 ura3-1 his3-10 1trp1-1 leu2-112 can1-100*/pNOY102	[33]
∆*rpa135*/pDK16	*MATa rpa135::LEU2 ade2-l ura3-l his3-ll trpl-1 leu2-3,112 can1-100*/pDK16	This study
YLL53	*MATa ade2-101 his3*∆*200 ura3-52**tyr1**RRP2*	[23]
YLL54	*MATa ade2-101 his3*∆200 *ura3-52*, *lys2*, *rrp2-2*	[23]
pDK16	YEplac112 carrying the rRNA transcription unit expressed from the CUP1 promoter	[35]
pDK16-tag	pDK16 with tagged 5,8S gene (see text)	This study

**Table 3 ijms-22-06690-t003:** Summary of characteristics of ITS1 deletions.

			ITS1 Nucleotides				L-Fraction			
Mutant	A2	A3	283–289	290–299	300–330	331–340	∆*rpa12* ∆*xrn1*	*RRP2* (YLL53)	HI mw interm in ∆*rpa12* ∆*xrn1*	*RRP2* sensitivity
wt	yes	yes	P	P	P	P	0.21	0.20	−	+
∆13	no	yes	P	P	P	P	0.19	0.21	−	+
∆14	no	no	A	A	P	P	0.31	0.34	−	+
∆15	yes	no	A	A	P	P	0.26	0.40	−	+
∆16	yes	yes	P	P	P	P	0.22	0.14	−	+
∆17	yes	yes	P	P	A	A	0.61	0.88	+	−
∆18	yes	half	P	A	P	P	0.50	0.80	+	−
∆19	yes	no	A	A	A	P	0.56	0.64	−	nd
∆2	no	no	A	A	A	A	0.74	0.82	+	−

Columns 2 and 3: Yes/no indicates whether the A2 and/or A3 processing sites are present or not in the mutant rRNA gene. “Half “ indicates that only the upstream part of the processing site is present. Column 4 indicates whether the indicated range of ITS1 nucleotides are present (P) or absent (A) in each mutant. Column 5 shows the L-fraction (5.8S_L_/(5.8S_S_+5.8S_L_) in the indicated strain. Column 6 indicates whether or not high molecular weight intermediates accumulate during ITS1 processing in each mutant (+: high mw intermediates accumulate; −: high mw intermediates do not accumulate). Column 7 indicates whether the L-fraction is significantly increased in the *RRP2* strain versus the *rrp2-2* mutant strain).

## Data Availability

Not applicable.

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
