# Peer review of "A Novel Model for the RNase MRP-Induced Switch between the Formation of Different Forms of 5.8S rRNA"

_ijms, 2021, doi:10.3390/ijms22136690_

Round 1

Reviewer 1 Report

In this manuscript, Li et al. apply an elegant pre-ribosomal RNA reporter system to characterise ITS1 spacer removal in budding yeast.

Although similar approaches have been utilised before, the findings in this report reveal fresh insights into the pathways that generate the long and short forms of the 5.8S rRNA.  Indeed, data presented here support the existence of a novel minor pathway to generate the long form of the 5.8S rRNA, which bypasses canonical endonucleolytic cleavages at ITS1 sites A2 and A3, and instead relies on D-site cleavage at the 3’-end of 18S followed by Xrn1-mediated 5’-3’ exonucleolytic digestion.

While it could be criticised that this novel pathway may only happen on artificial pre-rRNA constructs (such as the Δ2 construct used here), it becomes clear in the discussion that the authors had already observed a similar phenotype in a previous study (REF 24 : “Lindahl, et al., RNase MRP is required for entry of 35S precursor rRNA into the canonical processing pathway, RNA, 2009”), which investigated the processing of natural pre-ribosomal RNA in a yeast strain expressing a temperature sensitive RNase MRP mutant.

Overall, all experiments are well-performed and largely support the authors’ conclusions (see comments below). However, to help the reader to appreciate the significance of the new results (i.e. the existence of a new ITS1 removal pathway) also in the context of “normal” pre-rRNA processing, the authors previous observations described in REF 24 (see above) should be included in the introduction.

In addition, while it is clear from the results presented here that the MRP-induced switch between long and short 5.8S rRNA is not dependent on A3 site cleavage and does even occur in the absence of site A3, it seems very speculative to propose, as stated in the abstract, that this is due to “RNase MRP processing RNA molecules other than pre-rRNA”. 

Further experiments would be needed to support this statement, which, at the current point, should likely be rephrased to say that the link between RNase MRP activity and 5.8S 5’-end formation is currently not fully understood and that the effect seen upon expression of RNase MRP mutants is likely indirect.

Minor points that should be addressed prior to publication:

Introduction:

Page 1, line 51:

“The canonical model for processing of the primary Pol I transcript begins with the Utp24 and Rnt1 endonucleases splitting the 5’ ETS and 3’ ETS, respectively, from the main portion of the pre-rRNA, generating the 32S intermediate (Figure 1B) [12-14]”

  • This sentence, as well as the depiction of the processing pathway in Figure 1, seems to suggest that Utp24 is responsible for cleavages at sites A0 AND A1 in the 5’ ETS. While it is correct that Utp24 has been identified as the A1 site endonuclease, it is currently unclear which enzyme cleaves at site A0. Please rephrase and amend the figure.

It would also be good to include further enzymes into Figure 1, i.e. Las1 (endonucleolytic cleavage at site C2), Rat1 (processing of the 5’-end of 25S) and a question mark next to the endonuclease proposed to cleave at site B1L.

Page 4, line 58:

  • Please add Lygerou et al., Science 1996 (REF 40) to indicate the original identification of RNase MRP is the A3 site endonuclease.

Results: 

Figure 2:

  • It would be helpful to include the exact nucleotide position of the A2 and A3 cleavage sites in panel A.
  • Statistical analysis should be applied when calculating 5.8S L/S ratios in strains expressing the different constructs (here and in Figure 4) – how many biological repeats have been performed? Error bars?

Figure 3:

  • This figure could be improved be providing an enlarged insert around the 5’ end(s) of 5.8S showing the identity of the nucleotides in the sequence ladder.

Page 10, line 171:

  • Please provide more information about the rrp2-2 mutant.

Page 19, line 181:

“These results corroborate the notion that ITS1 sequences downstream of A3 contain the most important determinants for the relative accumulation of long and short 5.8S rRNA”

  • However, the observed differences between mutant Δ13 and Δ16, as well as between Δ14 and Δ15, respectively, also suggest that the presence of the A2 site can have an impact. Please comment.

Figure 4:

  • Please provide L/S ratios and statistical analyses for the different strains. It might be helpful to plot them as a graph to highlight the differences between WT MRP and the rrp2-2 mutant.
  • It would be good to have a loading control.

Page 12, line 209:

“stream of the proper D site (Figure 5, lane 4, “D-~10”). (A lesser amount of the “D-~10” band is generated in Δ13 (Figure 5, lane 6), as well as the canonical D band.)”

  • This should be lane 6 first, then lane 4.
  • Interestingly, a similar ectopic or “aberrant” cleavage at position -10 has previously been observed in in vitro cleavage experiments with yeast Nob1 (see Pertschy et al., JBC, 2009), especially when the hairpin surrounding the D-site has been disrupted.

Figure 6:

  • Panels A/ B: It would be good to include an agarose northern blot to compare the phenotypes seen on long precursors (i.e. 24S) in the absence and presence of Xrn1.
  • Panel B: What are the 5’-ends of fragments X and X’? Do these correspond to the cleavages at D-site and D-10, respectively? A primer extension experiment similar to the one presented in Figure 5 would be useful.

Figure 7:

  • To better judge the positions of (pre-)40S and (pre-)60S complexes on the gradient (in order to understand the sedimentation behaviour of 24S), the position of 18S and 25S mature rRNAs should be determined.
  • Please mark the position of the fragments belonging to the “exoladder of misprocessed 18S”, as seen in Figure 6, if possible?

Discussion:

Page 24, line 454:

“Previous base substitution experiments suggested that base-pairing of ITS1 sequences with nucleotides in the distal part of 18S rRNA block access for the Nob1 nuclease to the D site until A2 has been cleaved [55].”

  • This model has been disputed in later studies, including Lebaron et al., NSMB, 2012.

Author Response

Reviewer 1

We thank the reviewer for spending time and effort reviewing our manuscript and providing valuable suggestions for improvements of our manuscript.

In this manuscript, Li et al. apply an elegant pre-ribosomal RNA reporter system to characterise ITS1 spacer removal in budding yeast.

Although similar approaches have been utilised before, the findings in this report reveal fresh insights into the pathways that generate the long and short forms of the 5.8S rRNA.  Indeed, data presented here support the existence of a novel minor pathway to generate the long form of the 5.8S rRNA, which bypasses canonical endonucleolytic cleavages at ITS1 sites A2 and A3, and instead relies on D-site cleavage at the 3’-end of 18S followed by Xrn1-mediated 5’-3’ exonucleolytic digestion.

While it could be criticised that this novel pathway may only happen on artificial pre-rRNA constructs (such as the Δ2 construct used here), it becomes clear in the discussion that the authors had already observed a similar phenotype in a previous study (REF 24 : “Lindahl, et al., RNase MRP is required for entry of 35S precursor rRNA into the canonical processing pathway, RNA, 2009”), which investigated the processing of natural pre-ribosomal RNA in a yeast strain expressing a temperature sensitive RNase MRP mutant.

Overall, all experiments are well-performed and largely support the authors’ conclusions (see comments below). However, to help the reader to appreciate the significance of the new results (i.e. the existence of a new ITS1 removal pathway) also in the context of “normal” pre-rRNA processing, the authors previous observations described in REF 24 (see above) should be included in the introduction.

We have changed the Introduction to better express the significance of our findings and “catch the interest” of potential readers.

In addition, while it is clear from the results presented here that the MRP-induced switch between long and short 5.8S rRNA is not dependent on A3 site cleavage and does even occur in the absence of site A3, it seems very speculative to propose, as stated in the abstract, that this is due to “RNase MRP processing RNA molecules other than pre-rRNA”. 

Further experiments would be needed to support this statement, which, at the current point, should likely be rephrased to say that the link between RNase MRP activity and 5.8S 5’-end formation is currently not fully understood and that the effect seen upon expression of RNase MRP mutants is likely indirect.

We have reworded the end of the Abstract while maintaining that the link almost certainly must involve RNase MRP cleavage in a yet unknown target. This follows from the fact that RNase MRP’s only know activity is RNA cleavage.

Minor points that should be addressed prior to publication:

Introduction:

Page 1, line 51:

“The canonical model for processing of the primary Pol I transcript begins with the Utp24 and Rnt1 endonucleases splitting the 5’ ETS and 3’ ETS, respectively, from the main portion of the pre-rRNA, generating the 32S intermediate (Figure 1B) [12-14]”

  • This sentence, as well as the depiction of the processing pathway in Figure 1, seems to suggest that Utp24 is responsible for cleavages at sites A0 AND A1 in the 5’ ETS. While it is correct that Utp24 has been identified as the A1 site endonuclease, it is currently unclear which enzyme cleaves at site A0. Please rephrase and amend the figure.

We have modified Fig 1 and the text accordingly.

It would also be good to include further enzymes into Figure 1, i.e. Las1 (endonucleolytic cleavage at site C2), Rat1 (processing of the 5’-end of 25S) and a question mark next to the endonuclease proposed to cleave at site B1L.

Done.

Page 4, line 58:

  • Please add Lygerou et al., Science 1996 (REF 40) to indicate the original identification of RNase MRP is the A3 site endonuclease.

 Done.

Results: 

Figure 2:

  • It would be helpful to include the exact nucleotide position of the A2 and A3 cleavage sites in panel A.

We do not understand this comment. The exact cleavage sites were already indicated with arrows in our first submission.

  • Statistical analysis should be applied when calculating 5.8S L/S ratios in strains expressing the different constructs (here and in Figure 4) – how many biological repeats have been performed? Error bars?

We have quantified all bands in the autoradiograms and calculated appropriate statistical parameters. The results are given in a new figure (Figure 5) and Table S1.

Figure 3:

  • This figure could be improved be providing an enlarged insert around the 5’ end(s) of 5.8S showing the identity of the nucleotides in the sequence ladder.

Done.

Page 10, line 171:

  • Please provide more information about the rrp2-2 mutant.

We have expanded our comments on the two RRP2 mutants (rrp2-2 and Mini2) and the phenotypic difference between them.

Page 19, line 181:

“These results corroborate the notion that ITS1 sequences downstream of A3 contain the most important determinants for the relative accumulation of long and short 5.8S rRNA”

  • However, the observed differences between mutant Δ13 and Δ16, as well as between Δ14 and Δ15, respectively, also suggest that the presence of the A2 site can have an impact. Please comment.

We have modified the text to indicate that although the mutants with the highest L/S (now L-fraction = L/(L+S)) include mutations in the distal ITS1 region, loss of the A3 site (∆14 and ∆15) also impacts the L/S ratio (now L-fraction=L/(L+S)). However, comparing ∆14 with ∆15 and ∆13 with ∆16 does not point to an importance of the A2 site in determining the L-fraction. Surely, more work is necessary to understand the dynamics and regulation of the ITS1 processing, but such work is beyond the scope of this manuscript.

Figure 4:

  • Please provide L/S ratios and statistical analyses for the different strains. It might be helpful to plot them as a graph to highlight the differences between WT MRP and the rrp2-2 mutant.

As indicated above, we now include quantitative and statistical analyses graphed in Figure 5.

  • It would be good to have a loading control.

Most of our arguments depend in comparisons within each lane. However, we have provided a loading control (O20 showing the sum of all 5.8S) in Figure 7 (previously Figure 6).

Page 12, line 209:

“stream of the proper D site (Figure 5, lane 4, “D-~10”). (A lesser amount of the “D-~10” band is generated in Δ13 (Figure 5, lane 6), as well as the canonical D band.)”

  • This should be lane 6 first, then lane 4.

Thank you for pointing this out.

  • Interestingly, a similar ectopic or “aberrant” cleavage at position -10 has previously been observed in in vitro cleavage experiments with yeast Nob1 (see Pertschy et al., JBC, 2009), especially when the hairpin surrounding the D-site has been disrupted.

We have included a comment and reference to the in vitro results of Pertschy et al.

Figure 6:

  • Panels A/ B: It would be good to include an agarose northern blot to compare the phenotypes seen on long precursors (i.e. 24S) in the absence and presence of Xrn1.

We agree, but for logistical reason it is unfortunately not possible for us to perform additional experiments.

  • Panel B: What are the 5’-ends of fragments X and X’? Do these correspond to the cleavages at D-site and D-10, respectively? A primer extension experiment similar to the one presented in Figure 5 would be useful.

The primer extension in Figure 5 (now Figure 6) was performed with O9, positioned immediately downstream of O453 used to identify the X fragment. The 5’ end of X must thus almost certainly be the D site. Since X’ is not seen with O453, it must have a 5’ downstream of ITS1 position 25, but not much downstream, since the electrophoretic mobility of X and X’ differs little.  We agree that it would have been good to do an additional primer extension with a primer downstream of O9. Overall, we do not think that the lack of mapping the 5’ end of X’ affects the conclusions of our work.

Figure 7:

  • To better judge the positions of (pre-)40S and (pre-)60S complexes on the gradient (in order to understand the sedimentation behaviour of 24S), the position of 18S and 25S mature rRNAs should be determined.

As indicated above, additional experiments are not possible. We have quantified the 5.8S bands across the gradient (plotted in the revised figure, now Figure 8). This shows that 24S-containing particle(s) sediment significantly slower than (pre-)60S complexes. In our judgement, at least some of the 24S complexes sediment slower than 40S, a region rich in transcribed spacer sequences (Gregory and Lindahl, unpublished).

  • Please mark the position of the fragments belonging to the “exoladder of misprocessed 18S”, as seen in Figure 6, if possible?

Done.

Discussion:

Page 24, line 454:

“Previous base substitution experiments suggested that base-pairing of ITS1 sequences with nucleotides in the distal part of 18S rRNA block access for the Nob1 nuclease to the D site until A2 has been cleaved [55].”

  • This model has been disputed in later studies, including Lebaron et al., NSMB, 2012.

Sorry for overlooking this important critique of the Lamanna model. We have removed the discussion of the model.

Reviewer 2 Report

In this manuscript Li, Zengel, and Lindahl investigate the role of the ITS1 spacer in generating the end of the 5.8S rRNA. There are two forms of the 5.8S (long and short) that are generated by two different pre-rRNA processing pathways. One pathway utilizes RNase MRP and the other an unknown endonuclease. To study the ITS1 the authors used a clever genetic strategy to look at pre-rRNA transcription and processing via a plasmid in a Pol I inactivated strain of yeast. Multiple deletions were made to the ITS1 and the authors found that deletions downstream of the A3 cleavage site alter the ratio of the long and short form of the 5.8S. The authors found that mutations in RNase MRP also impact the ratio of long and short even when the RNase MRP cleavage site is deleted. This finding suggests that RNase MRP is an import switch for controlling ITS1 processing.

Major Concerns:

  • This manuscript is lacking any description of reproducibility. All the results shown in the figures represent single experiments. No replicates are shown or mentioned and for measured values such as the ratio of L/S of the 5.8S there is no error/standard deviation. Moreover, there are no loading controls on any of the northern blots.
  • Please add sufficient detail to the “4.3 Other procedures” section of the methods so that someone reading the manuscript can understand the basic methods without having to look up previous work.
  • Identification of a strain of yeast that exclusively produces the long-form of the 5.8S is an exciting result, however the authors do not emphasize this result and instead focus on the impacts of other RNA processing factors (RNase MRP, Xrn1) in impacting the L/S ratio. The novelty/impact of this manuscript would be great increased if the authors characterized the delta 2 strain to determine what impact only having the long form of the 5.8S has on cells. For example, are there any changes in translation? Is this strain more sensitive to stress?
  • How do deletions of the ITS1 impact the ribosome assembly pathway? All northern blots were run and cropped to focused on the end of the 5.8S. Are there visible defects in other pre-rRNA intermediates upstream of ITS1 processing or the total RNA (25S and 18S)?
  • Please add a supplemental figure showing the growth curves/plates for the ITS1 deletions described in Fig. 2.
  • Why is the northern blot in Fig. 2H longer and contain an additional higher band labeled VL. Please describe this in the figure legend.
  • On lane 171 please include more information about the RNase MRP strain. What is the specific mutation in rrp2-2?
  • Fig 3 appears to be two different blots cropped together. The bands for the primer extension are significantly larger/wider than those of the ladder. Please explain/clarify how this was created in the figure legend. It’s also very difficult to see the * on the gel.
  • Please determine the L/S ratio for all the lanes in Fig. 4
  • In Fig. 5 why is only a signal lane from the ladder shown (G lane)?
  • Fig. 7 is missing the actual chromatogram/trace from the sucrose gradient. This must be included so that one can see if there are defects in ribosome production and/or polysome formation.

Minor Points:

  • Text in lines 223-229 and 306-324 appears to be a different size/font
  • The discussion is quite wordy and would be more effective if it was trimmed.

Author Response

Reviewer 2

We thank the reviewer for spending time and effort reviewing our manuscript and providing valuable suggestions for improvements of our manuscript.

In this manuscript Li, Zengel, and Lindahl investigate the role of the ITS1 spacer in generating the end of the 5.8S rRNA. There are two forms of the 5.8S (long and short) that are generated by two different pre-rRNA processing pathways. One pathway utilizes RNase MRP and the other an unknown endonuclease. To study the ITS1 the authors used a clever genetic strategy to look at pre-rRNA transcription and processing via a plasmid in a Pol I inactivated strain of yeast. Multiple deletions were made to the ITS1 and the authors found that deletions downstream of the A3 cleavage site alter the ratio of the long and short form of the 5.8S. The authors found that mutations in RNase MRP also impact the ratio of long and short even when the RNase MRP cleavage site is deleted. This finding suggests that RNase MRP is an import switch for controlling ITS1 processing.

Major Concerns:

  • This manuscript is lacking any description of reproducibility. All the results shown in the figures represent single experiments. No replicates are shown or mentioned and for measured values such as the ratio of L/S of the 5.8S there is no error/standard deviation.

We have added a section on the quantification of the of the L/S ratio (we now express the variation in the production of the two 5.8S forms as L/L+S, because the simple L/S ratio is more sensitive to variations in the abundance of the S form in strains making little 5.8SS). In Materials and Methods, we now also point out that all “observations” are supported by at least two experiments, although the experimental design may differ.

Moreover, there are no loading controls on any of the northern blots.

Our arguments depend in comparisons within each lane. In Figure 7 (Figure 6 in the first submission) where comparison between lanes is important, we have provided a loading control, see the probing with O20 (showing the sum of all 5.8S) and note that lanes 1-5 and 6-10 is the same blot probed sequentially with O552, O576, and O20.

  • Please add sufficient detail to the “4.3 Other procedures” section of the methods so that someone reading the manuscript can understand the basic methods without having to look up previous work.

We have added brief descriptions of methods in section 4.3

  • Identification of a strain of yeast that exclusively produces the long-form of the 5.8S is an exciting result, however the authors do not emphasize this result and instead focus on the impacts of other RNA processing factors (RNase MRP, Xrn1) in impacting the L/S ratio. The novelty/impact of this manuscript would be great increased if the authors characterized the delta 2 strain to determine what impact only having the long form of the 5.8S has on cells. For example, are there any changes in translation? Is this strain more sensitive to stress?

Although we agree that these are interesting questions, the investigation of such matters is beyond the scope of this manuscript.

  • How do deletions of the ITS1 impact the ribosome assembly pathway? All northern blots were run and cropped to focused on the end of the 5.8S. Are there visible defects in other pre-rRNA intermediates upstream of ITS1 processing or the total RNA (25S and 18S)?

The purpose of our study was to understand the processing in ITS1 and the distribution of long and short 5.8S rRNA. However, we have also pointed out aberrant cleavages in the 18S rRNA seen in the processing of ∆2 pre-rRNA (Fig 6).

  • Please add a supplemental figure showing the growth curves/plates for the ITS1 deletions described in Fig. 2.

Figure S2 now shows plates of ∆2, ∆4, and ∆7 growth with controls. Unfortunately, we did not take pictures of plates showing the growth of the other mutants. However, comparisons between growth rates or colony sizes of the different strains can be misleading, since the rRNA genes are located on plasmids whose copy number may vary between strains, since there will be a selection for cells with the highest copy numbers. (There is not a mechanism assuring equal division of plasmids like the chromosomal segregation). Even if we had integrated the mutant rRNA genes in the chromosome, there would likely be a selection for high copy number due to recombination within the chromosomal array. Thus, we have focused on pathways rather than cell physiology.

  • Why is the northern blot in Fig. 2H longer and contain an additional higher band labeled VL. Please describe this in the figure legend.

We have removed northern blots from Figure 2; northern blots of the 5.8S region in all strains are now presented in Figure 4. Furthermore, we inserted a new Figure S2 showing accumulation of larger processing intermediates in the processing of some mutants in the Xrn1 background and added corresponding text.

  • On lane 171 please include more information about the RNase MRP strain. What is the specific mutation in rrp2-2?

We have expanded our comments on the two RRP2 mutants (rrp2-2 and Mini2) and the phenotypic difference between them.

  • Fig 3 appears to be two different blots cropped together. The bands for the primer extension are significantly larger/wider than those of the ladder. Please explain/clarify how this was created in the figure legend. It’s also very difficult to see the * on the gel.

This issue is now clarified in the legend to the figure (Now Figure 4).

  • Please determine the L/S ratio for all the lanes in Fig. 4

We have now quantified northern blots for all ITS1 mutants. The results are given in Figure 5 and Table S1.

  • In Fig. 5 why is only a signal lane from the ladder shown (G lane)?

The products of a single di-deoxy reaction provide sufficient markers if the sequence is already known. The method is regularly seen in the literature. To facilitate the interpretation of the g-lane we have inserted the sequence around the D-site in the figure.

  •  
  • Fig. 7 is missing the actual chromatogram/trace from the sucrose gradient. This must be included so that one can see if there are defects in ribosome production and/or polysome formation.

Unfortunately, we do not have the trace, and for logistical reasons we cannot do more experiments. However, we have quantified the 5.8S in fractions across the gel. The point we make from this experiment is that the 5.8S 5’ extensions are not (fully) removed before the molecule is inserted into ribosomes.

Minor Points:

  • Text in lines 223-229 and 306-324 appears to be a different size/font

Corrected

  • The discussion is quite wordy and would be more effective if it was trimmed.

We have attempted to make the text more succinct.

Round 2

Reviewer 2 Report

The authors have adequately addressed most of my previous concerns. The additional data analysis of the ratio of long/short 5.8S is a nice addition to the manuscript.